# What works best for ensuring treatment adherence. Lessons from a social support program for people treated for tuberculosis in Ukraine

Zulfiya Charyeva[1,2]*, Siân Curtis[1,3◉], Stephanie Mullen[1,4◉], Tatyana Senik[5‡], Olga Zaliznyak[5‡]

**1** MEASURE Evaluation Project, Carolina Population Center, University of North Carolina at Chapel Hill, Chapel Hill, North Carolina, United States of America, **2** Palladium, Chapel Hill, North Carolina, United States of America, **3** Department of Maternal and Child Health, Gillings School of Global Public Health, University of North Carolina at Chapel Hill, Chapel Hill, North Carolina, United States of America, **4** John Snow Inc., Rosslyn, Virginia, United States of America, **5** International Research Agency IFAK Institut, Kyiv, Ukraine

◉ These authors contributed equally to this work.
‡ These authors also contributed equally to this work.
* Zulfiya.Charyeva@thepalladiumgroup.com

**Data Availability Statement:** Data from the Ukraine Strengthening Tuberculosis Control Project Impact Evaluation – Phase 2 are available online

## Abstract

### Background

Worldwide, TB is one of the top 10 causes of death and the leading cause from a single infectious agent. Ukraine is one of 30 countries with the highest burden of multidrug-resistant tuberculosis. Global literature shows that social support (SS) is important in improving TB treatment adherence, reducing lost to follow up rates and improving treatment outcomes. There are several models of SS available, and the literature provides little information on what aspects of SS are most important to TB patients in improving their adherence.

### Methods

We used qualitative data collected through in-depth interviews (IDI) with 21 TB patients and 15 SS providers and coordinators in Ukraine in August-September 2016 to understand how the SS program promoted treatment adherence among patients. We examined the aspects of outpatient TB treatment that made adherence particularly difficult for patients in at-risk groups and aspects of the SS programs that worked best for addressing those barriers. Interviews were transcribed and analysis was performed to derive emergent themes.

### Results

Main barriers included side effects from medicine, the amount of time required daily for transportation and waiting in lines at the health facility, transportation expenses, risks of being identified when visiting a TB facility and lack of motivation to seek treatment. Features of the SS program most valued by patients were convenience of not having to visit facility and support provided by nurses. These two features directly addressed most of the barriers

from the University of North Carolina, Odum Dataverse repository (https://dataverse.unc.edu/dataset.xhtml?persistentId=doi:10.15139/S3/MUSYTQ) published in MEASURE Evaluation Dataverse (view at https://dataverse.unc.edu/dataverse/MEP3). Anonymized transcripts are uploaded to the repository (https://dataverse.unc.edu/dataset.xhtml?persistentId=doi:10.15139/S3/MUSYTQ) and are available to researchers at no charge.

**Funding:** This publication was produced with the support of the United States Agency for International Development (USAID) under the terms of MEASURE Evaluation cooperative agreement AID-OAA-L-14-00004. MEASURE Evaluation is implemented by the Carolina Population Center, University of North Carolina at Chapel Hill in partnership with ICF International; John Snow, Inc.; Management Sciences for Health; Palladium; and Tulane University. Views expressed are not necessarily those of USAID or the United States government. The funder had no role in study design, data collection and analysis, decision to publish, or preparation of the manuscript.

**Competing interests:** The authors have declared that no competing interests exist.

identified. The commitment and qualities of the nurses that provided the SS was an important element of the program.

## Conclusions

This qualitative study suggests that the SS program in Ukraine was successful in reducing treatment default among patients at high risk of default because it directly addressed most of the major barriers they faced to treatment adherence.

## Introduction

Ukraine is one of 30 countries with the highest burden of multidrug-resistant tuberculosis (MDR TB) [1]. It had an estimated 20,000 new cases of multidrug-resistant or rifampicin-resistant TB (MDR/RR-TB) in 2017 [1]. Out of 30 countries with a high MDR-TB burden, only four, including Ukraine, had 28 and higher percentage of new TB cases with MDR/RR-TB in 2017 [1]. MDR-TB is associated with poor treatment adherence. [2] Global literature shows that social support (SS) is important in improving TB treatment adherence [3, 4], reducing lost to follow up rates [5, 6] as well as improving successful treatment outcomes [7]. A systematic review of studies published between 2000 and 2017 [5] found that cohorts that received any form of psychosocial or material support had lower lost to follow up rates during drug resistant TB treatment than those that received standard care. Meta-analysis conducted by Hoorn and colleagues [7] showed that psycho-emotional, socio-economic and combination of these types of support provided to patients with TB were associated with a significant improvement of successful treatment outcomes.

To improve treatment adherence and subsequent treatment outcomes among populations at high risk for treatment default in Ukraine, USAID supported the home-visiting social support program for TB patients vulnerable to treatment default, implemented by the Ukrainian Red Cross Society (URCS) under the Strengthening Tuberculosis Control in Ukraine (STbCU) program grant. Daily home visits by nurses provided delivery and direct observation of treatment, along with information materials to encourage full TB treatment adherence. Informational campaigns were conducted to increase TB knowledge in the society. The social support program targeted ten high risk (HR) group patients for treatment default: HIV-positive, alcoholics, people who inject drugs, TB contacts, homeless, migrants, refugees, ex-prisoners, unemployed, persons with comorbidities, and others identified as HR by the health care provider. An impact evaluation of the social support program established that participation in the SS intervention improved TB treatment outcomes among TB patients at high risk of default. The intervention cohort had higher treatment success and lower likelihood of treatment default and dying than the other two high risk patients' comparison groups [8]. However, there are several models of SS available including counseling by health care workers, peer support, delivering services, financial assistance, with different associated costs, and the literature provides little information on what aspects of SS are most important to TB patients in improving their adherence.

The objective of this study—conducted by MEASURE Evaluation, a project funded by USAID and the United States Presiden't Emergency Plan for AIDS relief—is to better understand how and why the SS program in Ukraine supported adherence among TB patients at high risk of default. Findings from this study will add to the evidence base for TB strategies and be useful to various stakeholders in Ukraine and internationally to guide decision making

about the social support program design and implementation. We use qualitative data collected through in-depth interviews (IDI) with TB patients and SS providers in Ukraine to understand how the social support program promoted treatment adherence among patients. In particular, we examined the aspects of outpatient TB treatment that made adherence particularly difficult for patients in at-risk groups and aspects of the social support programs that worked best for addressing those barriers.

## Methods

### Participant selection and data collection timeframe

Patient and provider interviews were completed with patients receiving and nurses providing URCS services in Ukraine in 2016. We interviewed respondents in two regions (Odessa and Dnipropetrovsk) where the quantitative impact evaluation of the social support program on treatment outcomes took place and where URCS was still providing SS services to patients. In-depth interview respondents included both male and female patients to examine potential differences in barriers to treatment adherence by sex, and the means of overcoming those barriers. We asked each URCS office to provide a list of nurses who worked in the SS program and then contacted eleven of those nurses to interview using convenience sampling. Nurses nominated their patients for interviews. Nominated patients who had received home visits for at least two months or those patients who had completed the program no longer than two months previously were invited for interviews. Program coordinator interviews were completed with the STbCU and URCS managers working on the SS program in both regions and in Kiev. All interviews were conducted in August and September 2016.

### Data collection procedures

We developed tailored, semi-structured interview guides for program beneficiaries, providers, and program coordinators. All guides were translated into Ukrainian and Russian. Interviews were conducted in both languages, depending on the preference of the respondents. The guides were pre-tested with two patients and one nurse in Dnipropetrovsk, and minor changes were made to improve the clarity and intent of the questions.

We used patient, provider, and STbCU staff interviews to gather in-depth information on what services were provided, who was using those services and how, and to what extent services in the delivery models were working for the intended audience. To better understand the role of SS services in treatment adherence, in-depth patient interviews solicited information from HR patients on the primary barriers to treatment adherence and aspects of the SS program that helped them stay on their treatment regimen. We interviewed STbCU staff and URCS coordinators to learn about their experiences coordinating the SS program; specifically, the barriers to and facilitators of their work, and lessons learned that can be applied to future programs.

We informed participants of the study aims, risks and benefits for participation in the study, and obtained verbal informed consent prior to the interview. We conducted interviews with program beneficiaries in parks or in a private and quiet location in the local URCS offices, out of earshot of program staff. Interviews with nurses were conducted in their places of work. Interviews with the STbCU staff and URCS program coordinators were conducted in their offices in Kiev and in the Odessa and Dnipropetrovsk regions. The interviews lasted approximately one hour. Interviews were audio-recorded using digital recorders, and a separate consent to record was sought by the interviewers. All study protocols, consent forms, tools, and data security processes were reviewed and approved by the Institutional Review Board at UNC-CH and the ethics review board at the F.H. Yanovskyi Institute of Phthisiology and Pulmonology under the Academy of Medical Sciences of Ukraine.

## Data analysis

All interviews were transcribed and then translated into English. Transcripts were imported into ATLAS.ti, version 7.5.17 and analyzed. Study staff developed an initial codebook with topical codes based on questions from the interview guides. The codebook was then pilot tested on interview transcripts for two patients (one from each region) and two providers (one from each region). The pilot testing allowed for the revision of the codebook; new codes were added, and some initial codes were collapsed into existing codes.

Once the codebook was finalized, the transcribed interview files were imported into ATLAS.ti to facilitate analysis, and the codes from the revised codebook were applied to the interview transcripts. Once coding was completed, a code report was run in ATLAS.ti for each code across each stakeholder group (patient interviews, provider interviews, project coordinator interviews). Two researchers read all study transcripts and reviewed the code reports, identified sub-themes in each code, examined the evidence supporting the themes and sub-themes independently and then formed a consensus. Essential concepts and relationships between the different themes and sub-themes were formed. Data were synthesized, and findings communicated through the process of writing up and presenting the data, using direct quotes to support the themes.

## Results

### Study participants

We interviewed 21 patients and 11 social support program providers/nurses from four cities in two regions of Ukraine. Eight of the patients were female. All providers were female. Nine nurses worked full time for URCS and two nurses worked in a TB health facility (HF). We also interviewed one STbCU staff member and three URCS coordinators. See Table 1 for more information about the number of patients, nurses and coordinators interviewed by city and region.

### Study findings

We did not find any sex differences in barriers to adherence reported by patients. Therefore, the findings below are for all patients.

**Challenges to treatment adherence prior to the social support program.** We asked patients about the challenges to adherence they faced when they received outpatient TB treatment prior to their participation in the social support program. Major challenges were related to required daily visits to health facilities for directly observed treatment.

Most patients reported being too weak to walk and reported side effects from medicine such as nausea, dizziness, pain in joints, lack of memory and sleep disorders that prevented

**Table 1. Geographic distribution of the participants.**

| City | Region | Number of patients interviewed | Number of providers interviewed | Number of program coordinators |
|------|--------|-------------------------------|--------------------------------|-------------------------------|
| Odesa | Odesa | 10 | 5 | 1 |
| Dnipro | Dnipropetrovsk | 4 | 2 | 1 |
| Kryvyi Rih | Dnipropetrovsk | 5 | 2 | |
| Nikopol | Dnipropetrovsk | 2 | 2 | |
| Kyiv | | | | 2 |
| Total | | 21 | 11 | 4 |

them from being able to visit the HF daily. All patients interviewed relied on public transportation to get to a HF. It was challenging for patients to walk to a bus stop, wait for a bus, ride in a bus or a minivan full of people and then repeat the process going back home. One patient illustrated the challenges related to getting to health facilities using public transportation while having side effects from taking medicine:

*From the very beginning of treatment, I started having nausea and I was very sleepy. As I kept taking pills, my condition worsened. . . . It was a long way to a health facility. I had to wait for the minivan. I felt dizzy from the crowd in the minivan too. I felt weak and almost fainted from these pills. . . Sometimes I missed my stops when I was riding a minivan. I did not feel well. . . .. There were a few days when I could not get to health facility because I could not make myself get up and go. This was because of the side effects from pills. These are strong pills.*

*(male patient, Odesa)*

The amount of time required for transportation and waiting in lines at the HF presented challenges for the majority of the patients interviewed. Most patients spent anywhere from two to six hours daily traveling to the health facility, waiting in lines and returning home. As one patient stated, it took at least half of the day for her to do this daily.

*If I don't come in the morning, then I need to wait, maybe the nurse is busy with handing out medicine; it is inconvenient—you wait at the bus stop too, half a day goes by. It is hard.*

*(female patient, Dnipro)*

Those patients who spent two hours daily also felt that it was taking a lot of time they could use for something else. This patient described his challenges for daily visits:

*This (visits to HF) takes time. In my case, I have to walk to a tram stop for 15 minutes, then it takes time to get there (to HF), to take pills, to come back. At the end it takes about two hours. Time goes fast. However, there are other things in life that I have to do and places to go. I don't want to spend so much time for HF visits every day.*

*(male patient, Odesa)*

Patients had to pay transportation expenses to go to the HF and back daily. It was from two to four hryvnia (USD 0.08–0.16) one way but since most patients were unemployed, they could not afford such an expense. One patient illustrated how difficult it was to pay for transportation expenses by saying:

*If someone is unemployed and is the only bread winner in the family. . . how are they supposed to live if there are no money? He can't spend any money for a van or tram here and there. All he has is 5 hryvnia to buy some bread.*

*(male patient, Odesa)*

Patients reported dissatisfaction with health facility hours of operation. Health facilities administered treatment from 9 am to 5 pm, so those few patients who worked had to be there during working hours. This patient illustrated the inconvenience of the HF working hours:

*I can go to the health facility either in the evenings or before 9 am, before work. The health facility opens only at 9 am. And it is not convenient for me to go there during lunch time. It takes one hour to get there and then another hour for the way back.*

*(male patient, Odesa)*

In addition to the time spent waiting in line, long queues in HFs had some patients worried about the possibility of getting infected with another TB strain, leading to a desire for them to limit their visits to the HFs. While some HFs provided masks, others required patients to pay or bring their own, which patients could not afford. One patient expressed her fear of getting infected in a HF by stating:

*People are very different. One can't tell by looking at them. They sometimes cough with blood. It's horrible. If you have a passive form, you fear that you get infected with another form. . . This is scary, I have two children.*

*(female patient, Dnipro)*

Stigma was identified as one of the biggest challenges to patient adherence. All interviewed patients reported that they did not want to be seen in a TB health facility by their friends and acquaintances. Patients were hiding their disease from people, sometimes even relatives. Usually, there were only one or two TB facilities in a city. Patients said that sometimes people without TB visit these facilities for diagnosis. When they see a person they know standing in line to get pills or even on the way to or from the facility, they conclude that this person has TB and might spread this information to other people they know. Spreading this news could result in loss of social status and exclude the person from their circle of friends and acquaintances. One of the patients illustrated how stigma prevented him from visiting the health facility:

*Odesa is a very big communal apartment. It is like a very big village. . . . when somebody sees you in the hospital he says: "Why is he going to the tuberculosis dispensary?" Most conclude that probably he has tuberculosis. If you show up in the hospital it becomes clear that something is wrong with you. It goes without saying because you go to the tuberculosis dispensary. In our city this hospital specializes only in tuberculosis. There is one in every city, in every district. So it is important from the psychological point of view and public opinion matters. I really care about it.*

*(male patient, Odesa)*

One provider shed light on how stigma influence patients' quality of life:

*They also worry a lot. They go to TB dispensary and take their pills, but they live in constant fear that they will be seen there by someone. Drug users and alcoholics do not care whether they will be seen or not while normal people always worry.*

*(provider, Nikopol)*

Patients reported that some of the patients they know do not want to get treated, and, therefore, refuse taking pills:

*Half of the patients are very asocial. And they are not very interested in treatment. I know many people who do not want to receive any treatment and refuse medical aid whatsoever.*

*(female patient, Kryvyi Rih)*

**Benefits of the social support program that helped improve treatment adherence.** Two aspects of the Social Support program were stated as the most important to patients–home visits, since pills are brought to patients daily, and support provided by nurses.

The fact that pills were brought to a patient eliminated multiple adherence barriers related to getting to a HF to take pills such as transportation, time, lines in HF, inability to go to HF due to weakness, forgetting to go to take pills, etc. Thus, daily visits made adherence easier for patients. Patients described how visits made it simple for them to stay on treatment:

*Yes, there are no issues related with getting treatment. Everything is so simple that they bring it to you, you take it, and continue on with your activities.*

*(male patient, Nikopol)*

Almost all patients reported they appreciated not having to spend time, efforts and money to get to the HF daily and or waste time waiting in the lines at HFs.

*(The nurse) brings the pills to home, so that you don't have to go there, to be stuck in traffic. Even today you witnessed this (traffic). This is the norm for us. You will definitely not be on time, it is pure stress. This is not treatment, it is only stress.*

*(male patient, Odesa)*

Over half of patients reported that minimizing side effects helped them stay on treatment. Some appreciated the home visit program for the opportunity to take pills and go straight back to bed right afterwards to rest.

*After you take these pills, you want to lay down. She leaves, we say goodbye, I close the door and go lay down for a little. And you already feel a little better. Whenever you get nauseous, you want to lay down for a little.*

*(male patient, Odesa)*

Almost all nurses reported they encouraged their patients to take pills after a meal. About a third of patient participants mentioned that side effects were reduced when they took pills with or right after a meal, which was easier to do when patients were at home.

*Different effects occur. Meaning one day you take them and everything is ok but another day, you can become nauseous. As a rule, it depends on whether you have eaten. I noticed that if I take pills without food, then I may not feel well.*

*(male patient, Nikopol)*

Most patients stated that it was important for them to have treatment outside of TB facilities to minimize contact with other patients as well as acquaintances and neighbors visiting HFs for testing. This feature of the SS program addressed fears of getting infected with another TB strain while visiting the health facility. It also addressed the concerns related to the stigma and risk of social exclusion if patients were seen by others (friends, acquaintances, neighbors) during their HF visits.

*I feel comfortable now, not worrying about anything. When I go to the TB dispensary, which is in the city, I have a feeling that everyone looks at me. But here I am comfortable.*

*(male patient, Kryvyi Rih)*

All patients were happy with the flexibility of nurses in scheduling their meeting times. Patients reported that nurses made it a priority to meet the interests of patients and satisfy patient needs. Thus, they were willing to meet with patients in locations that worked for patients and at times that were convenient for them. In addition, nurses were very flexible in their schedule; they were willing to adjust their schedule depending on patient needs. Thus, if a patient could not meet at the agreed time, nurses found another time during the day that worked for patient's schedule.

*It is more convenient for me to meet with the nurse rather than go to the TB hospital. Everything works for me because it is a convenient time, a convenient location. We are always able to find a good fit. If I am unable to, then she will work around my schedule. I can also work around her schedule. That is what was most fitting for me—that you could always find a solution.*

*(female patient, Kryvyi Rih)*

*I believe that what works is satisfying the desired conditions of the patient—however he wants it, it will happen.*

*(provider, Kryvyi Rih)*

All patients appreciated the program for ensuring uninterrupted treatment. About half of patients said it would be more difficult for them to adhere to treatment but they would adhere and half said that they would stop treatment or have interruptions.

*Actually, when the nurse started to come I realized that it's better for me. If I had to go to the dispensary, I would have been going there for one week, but for another–no. And I would have never recovered.*

*(female patient, Kryvyi Rih)*

Patients appreciated that they did not have to remember to take pills, and that a nurse's daily visit served as a reminder.

*If the pills were in my home, then I think I could have forgotten about them. Yesterday I forgot about my vitamins and today, I also forgot to take them. . . . I know myself in regards to that I would not follow the correct schedule but with a nurse it is simple, it is always on time.*

*(female patient, Kryvyi Rih)*

From the interviews with nurses and patients, we learned that nurses possessed multiple qualities that promoted their work with patient adherence. Patients described nurses as open, sincere, approachable, responsible, flexible in scheduling, open for communication, being "positive", and having good energy. Nurses had excellent interpersonal communication skills. In particular, they emphasized the importance of active listening. Nurses called themselves "psychologists" since they had skills to earn patient trust and build rapport. All nurses emphasized that gaining a patient's trust was a door opener in their work with patients and patients also reported that they trusted their nurses. Nurses also stated the importance of treating patients as any other society member, as someone equal to them. They reported they accepted patients, understood their problems, challenges and needs. Patients valued and appreciated these attitudes toward them.

Nurses provided emotional, informational, instrumental (i.e. providing tangible assistance, offering a helping hand) and motivational support for their patients. All patients felt that nurses cared about them.

*Furthermore, she keeps calling me, never forgetting about doing medical tests: "Go to do X ray, go.". . .. I know that there is someone caring for me, who monitors my health and medical tests. . . I just really liked that she cares about me.*

*(male patient, Dnipro)*

They reported that nurses provided hope and a listening ear, and they showed respect and empathy.

*She (nurse) always asked "Name (omitted), how do you feel? How are you doing? Are you feeling well?" I say, "Yes, I am already getting better." She constantly asked "Do you take vitamins? Do you eat well?" I am not able to say that we are friends, but I can say that I felt a lot of support from her. When she came, I was not doing well. We spoke a lot. . . . She always told me: "You will get treated and everything will be alright, you'll find a job and everything will be normal." She supported me.*

*(female patient, Odesa)*

Patients reported they could trust nurses and could discuss any topic with them, including very personal issues:

*I was able to talk to her about anything. I could talk even about those things that I wouldn't have been able to talk about if I were standing in line.*

*(male patient, Odesa)*

Half of the interviewed patients reported feeling isolated from society. For these patients, their nurse was the only other person they could talk to.

*Tuberculosis is not a flu, but a sickness with which you need support. See, with another disease, you can talk to somebody, can share. With this disease, I cannot just talk to someone and pour my heart out. I was able to talk with the nurse, she knows.*

*(female patient, Nikopol)*

All interviewed patients reported that nurses continuously provided them and their family members with a lot of information such as facts about TB, side effects, importance of staying on treatment to prevent drug resistance, healthy nutrition, recipes, importance of exercise, walks, and personal hygiene. An example from a patient about the topics discussed:

*She explains as another human, shares such interesting things . . . For example, without her, the doctor said: "Take all your tablets, that is it." She explained it completely differently—she explained the necessity for taking pills. She explained that you do not die from TB.*

*(female patient, Dnipro)*

In addition to providing information verbally, nurses gave each patient a diary that contained information about tuberculosis, diet, and exercise. Patients were encouraged to fill it

out with any information about how they felt, record side effects, if any, record daily temperature, questions to ask a nurse, etc. Patients appreciated having this source of information, valued its content and simplicity and liked that it was easily accessible.

Patients reported that their nurses reminded them about upcoming doctor appointments or lab work, brought referrals for lab work and informed patients about lab results or a need to repeat lab work. Some nurses gave advice on the best time of the day to visit the facility to avoid lines. Some brought tickets/passes to patients for lab work for a specific day and time to minimize waiting time in a facility.

*She calls me and says: "You should visit the dispensary today. If you have time, you can go today, if no–you can go and get tested tomorrow. Skip the queue, say that you are referred by the nurse". That's why I always skip the queues. I enter the cabinet, doctor gives me the referral, I get tested and that's all.*

*(female patient, Kryvyi Rih)*

Nurses confirmed they brought referrals to patients for lab work and reminded patients about medical appointments. According to both patients and nurses, nurses were in continuous communication with patients' doctors. When patients had severe side effects, nurses communicated with a doctor and the doctor substituted treatment with other pills or prescribed medication to minimize side effects. Several nurses said that they even accompanied their patients to health facilities.

While home visits in the USAID-supported social support program did not have any incentives such as food or clothing, all nurses provided such support to patients using URCS resources or their own. Most program recipients did not work, or worked part time; as a result, they did not have means to buy food. Nurses reported that they felt that they had to support patients as much as they could. Thus, when food was available at URCS from sponsors and donors, nurses brought patients bread, dairy products, watermelons, and fruit. When clothes were available, nurses brought patients clothing from URCS or invited patients to URCS to try on and choose what they liked.

*(my nurse brought) a big can of beef. Based on my disease, as I understand, there are no food parcels. Of course, I also asked about receiving clothing. I have nothing to wear in the summer, no shoes, and no money to buy anything with. They looked, there is nothing available right now. If anything appears, then they will help me. They try to help me as much as they can.*

*(male patient, Nikopol)*

If food was not available at URCS, nurses bought bread and milk out of pocket to support their patients.

*From time to time, I buy some of my patients bread and milk. Simply I see the living conditions in which they live and try to help them in any way I can. They try to earn money wherever there is an opportunity... Sometimes I simply bring them a bottle of jam. I do not give them money because they can spend it on something else. In contrast, whenever I bring them food, I know that they will eat it and no longer be hungry.*

*(provider, Kryvyi Rih)*

There were a few instances when nurses loaned money to patients so they could buy food. Often nurses gave patients money for transportation expenses to get to the HF for lab work or a doctor's appointment.

*You know, they do not go to the medical facility. They came once or twice but then no longer had money. They do not come anymore . . . For this reason, I take her myself and accompany her. She doesn't have money to come on her own. Even if she did, she would not go on her own. I go with her there and support her, do you understand?*

*(provider, Odesa)*

In other instances, nurses reported bringing cleaning detergents to low income patients so they could keep their apartments clean.

Nurses encouraged patients to stay on treatment, motivated them and reassured patients that they had qualities to stay on treatment and complete it successfully.

*She only encourages me and says: "You are normal, you look fine, the results of the analyses are good. Soon this period of medicines intake will be over and you will be able to get official employment."*

*(male patient, Kryvyi Rih)*

## Discussion

In this study we identified outpatient treatment adherence barriers for patients at risk of treatment default in Ukraine, and described how the SS program worked to address most of the barriers. Main barriers included side effects from medicine, the amount of time required daily for transportation and waiting in lines at the health facility, transportation expenses, risks of being identified when visiting a TB facility and lack of motivation to seek treatment. Important features of the social support program valued by patients were convenience of not having to visit facility and support provided by nurses. These two features directly addressed most of the barriers identified which likely explains why the program was found to be successful in reducing treatment drop out among high risk patients participating in the program. While many of the barriers identified are likely to be generalizable to other TB patient populations, it is important for social support program designers to assess the specific barriers to treatment adherence faced by the TB patient population in their context and how the proposed social support program will address those barriers.

The qualities of the nurses and their excellent interpersonal communication skills appear to have been a critical component of the success of the program. One of the most important nurse qualities was that they really cared about patients and how they felt, and they sincerely wanted to help patients recover. This desire to help made nurses go above and beyond in providing services to patients. Nurses purchased food for patients out of pocket and gave them money for transportation to get to the health facility to do lab work, and patients treated nurses as their friends and family members. We may not see similar success in other contexts where nurses are not as highly motivated. If this social support program is to be replicated or scaled up, staff in SS programs in the future need to be trained to gain the trust of patients, build a close relationship with them, and have skills and qualities similar to those of nurses working for the URCS SS program. To prevent burn out and protect social support providers' emotional well-being, future programs may also need to provide skills on how to detach from patient problems after visiting patients.

The social support program was provided by the URCS and health facility nurses. We established that most of the social support program recipients were isolated from the society, felt lonely and valued the emotional and motivational support provided by nurses. It was very important for patients to have someone in their lives who cared about them. However, it is possible that to some extent patients followed nurses' recommendations and suggestions because they were aware of their medical education, working relationship with doctors and trusted their knowledge related to treatment regimen. Future studies would need to examine the effect of the social support program administered by other cadres such as social workers and provide recommendations on the potential for using these types of cadres to provide social support to high risk patients with TB.

To improve this type of SS program in the future, all groups of respondents suggested that food parcels or food certificates be offered to patients to support their treatment. When possible, employment and income generation opportunities should be provided as part of the intervention program. These recommendations are supported by existing research establishing that the economic burden for patients with TB is high and often includes income loss [9], and that economic support is important for improving treatment outcomes [7].

Stigma underlies several of the barriers to adherence that were identified. While the social support program reduces the impact of stigma on adherence by reducing the risks of being identified when visiting a TB facility, it does not address stigma at a structural level in society. Different types of programs are needed that aim to reduce the stigma in the larger society and reduce the isolation experienced by TB patients.

We have previously reported on the impact of the social support program on TB treatment outcomes using quantitative methods [8]. This qualitative study helped identify how the SS program in Ukraine supported adherence by addressing specific barriers to treatment adherence faced by the TB patients.

Our study had limitations. The sample was purposive and may not represent fully all experiences among high risk TB patients and providers. Patients were also recommended by providers and therefore, may have a positive bias towards the care they received through the SS program. We included various groups of respondents such as patients, providers and program coordinators to obtain perspectives from all groups involved in the social support program.

## Conclusion

This qualitative study suggests that the URCS social support program in Ukraine was successful in reducing treatment default among patients at high risk of default because it directly addressed most of the major barriers they faced to treatment adherence. The commitment and qualities of the nurses that provided the social support was an important element of the program. It is important for SS program designers to assess the specific barriers to treatment adherence faced by the TB patient population in their context and ensure that the proposed social support program will address those barriers. Future programs need to ensure they have high quality providers and consider including food parcels and other material support. Programs that address societal stigma against TB patients may also be needed in contexts where such stigma is widespread and is a barrier to treatment adherence.

## Acknowledgments

The authors would like to express sincere gratitude to the patients, nurses, the Ukrainian Red Cross Society program coordinators and the Strengthening Tuberculosis Control in Ukraine staff who shared their social support program experiences.

## Author Contributions

**Conceptualization:** Zulfiya Charyeva, Siân Curtis, Stephanie Mullen.

**Formal analysis:** Zulfiya Charyeva.

**Investigation:** Zulfiya Charyeva, Siân Curtis, Stephanie Mullen.

**Methodology:** Zulfiya Charyeva, Siân Curtis, Stephanie Mullen.

**Project administration:** Zulfiya Charyeva, Tatyana Senik, Olga Zaliznyak.

**Software:** Zulfiya Charyeva.

**Supervision:** Zulfiya Charyeva, Siân Curtis.

**Visualization:** Zulfiya Charyeva.

**Writing – original draft:** Zulfiya Charyeva.

**Writing – review & editing:** Siân Curtis, Stephanie Mullen, Tatyana Senik, Olga Zaliznyak.

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
