## [Decision Letter · Decision Letter 0]

5 Jul 2019

PONE-D-19-15537

What works best for ensuring treatment adherence. Lessons from a social support program for people treated for tuberculosis in Ukraine

PLOS ONE

Dear Dr. Charyeva,

Thank you for submitting your manuscript to PLOS ONE. After careful consideration, we feel that it has merit but does not fully meet PLOS ONE’s publication criteria as it currently stands. Therefore, we invite you to submit a revised version of the manuscript that addresses the points raised during the review process.

We would appreciate receiving your revised manuscript by Aug 19 2019 11:59PM. To enhance the reproducibility of your results, we recommend that if <gwmw class="ginger-module-highlighter-mistake-type-3" id="gwmw-15623349972774533662145">applicable you</gwmw> deposit your laboratory protocols in protocols<gwmw class="ginger-module-highlighter-mistake-type-3" id="gwmw-15623349972778159102839">.</gwmw><gwmw class="ginger-module-highlighter-mistake-type-1" id="gwmw-15623349972773470343332">io</gwmw>, where a protocol can be assigned its own identifier (DOI) such that it can be cited independently in the future. For <gwmw class="ginger-module-highlighter-mistake-type-3" id="gwmw-15623349977300200827305">instructions see</gwmw>: http://journals.plos.org/plosone/s/submission-guidelines#loc-laboratory-protocols

A rebuttal letter that responds to each point raised by the academic editor and reviewer(s). This letter should be uploaded as <gwmw class="ginger-module-highlighter-mistake-type-3" id="gwmw-15623349995292877521946">separate file</gwmw> and labeled 'Response to Reviewers'.A marked-up copy of your manuscript that highlights changes made to the original version. This file should be uploaded as <gwmw class="ginger-module-highlighter-mistake-type-3" id="gwmw-15623350010633380790531">separate file</gwmw> and labeled 'Revised Manuscript with Track Changes'.An unmarked version of your revised paper without <gwmw class="ginger-module-highlighter-mistake-type-3" id="gwmw-15623350017787276922391">tracked</gwmw> changes. This file should be uploaded as <gwmw class="ginger-module-highlighter-mistake-type-3" id="gwmw-15623350024957432010048">separate file</gwmw> and labeled 'Manuscript'.

Please <gwmw class="ginger-module-highlighter-mistake-type-3" id="gwmw-15623350035658574173647">note while</gwmw> forming your response, if your article is accepted, you may have the opportunity to make the peer review history publicly available. The record will include editor decision letters (with reviews) and your responses to reviewer comments. If eligible, we will contact you to opt in or out.

We look forward to receiving your revised manuscript.

Kind regards,

Kahabi Ganka Isangula, MD, MPH, PhD

Academic Editor

PLOS ONE

Journal Requirements:

2. Please provide additional details regarding participant consent. In the ethics statement in the Methods and online submission information, please ensure that you have specified (1) whether consent was informed and (2) what type you obtained (for instance, written or verbal). If your study included minors, state whether you obtained consent from parents or guardians. If the need for consent was waived by the ethics committee, please include this information

Reviewers' comments:

Reviewer's Responses to Questions

**Comments to the Author**

1. Is the manuscript technically sound, and do the data support the conclusions?

Reviewer #1: Yes

Reviewer #2: Yes

2. Has the statistical analysis been performed appropriately and rigorously? 

Reviewer #1: Yes

Reviewer #2: N/A

3. Have the authors made all data underlying the findings in their manuscript fully available?

Reviewer #1: Yes

Reviewer #2: Yes

4. Is the manuscript presented in an intelligible fashion and written in standard English?

Reviewer #1: Yes

Reviewer #2: Yes

5. Review Comments to the Author

Reviewer #1: A technically sound and academically well written manuscript. The qualitative data analysis was comprehensively explained. However, mention of inter rater reliability by comparing the coding and thematic analysis with co researchers and participants would have further strengthened the analysis. Nevertheless, the manuscript highlights relevant themes pertaining to adherence to TB treatment. Multiple drug resistant strains are a major public health concern of present times.

Reviewer #2: The manuscript is written in tangible and organized way and touched critical area that should be addressed. However, some printing errors need revision. e.g. line 196, line 270

6. PLOS authors have the option to publish the peer review history of their article (what does this mean?). If published, this will include your full peer review and any attached files.

Reviewer #1: No

Reviewer #2: No

<gdiv></gdiv>

---

## [Author Response · Author response to Decision Letter 0]

8 Aug 2019

We provided our response to comments in the file named "Response to reviewers." Our responses are below, followed by comments. 

Response to editors

We followed the instructions in the templates and made revisions in the manuscript to meet PLOC ONE’s style requirements. 

2. Please provide additional details regarding participant consent. In the ethics statement in the Methods and online submission information, please ensure that you have specified (1) whether consent was informed and (2) what type you obtained (for instance, written or verbal). If your study included minors, state whether you obtained consent from parents or guardians. If the need for consent was waived by the ethics committee, please include this information

We added details regarding participant consent (line 137-138 in the manuscript with tracked changes) and stated the following: “We informed participants of the study aims, risks and benefits for participation in the study, and obtained verbal informed consent prior to the interview.”

We de-identified all transcripts from in-depth interviews with study respondents and made them available online from the University of North Carolina, Odum Dataverse repository.

This is the Data Availability Statement: 

Data from the Ukraine Strengthening Tuberculosis Control Project Impact Evaluation – Phase 2 is available online from the University of North Carolina, Odum Dataverse repository (https://dataverse.unc.edu/dataset.xhtml?persistentId=doi:10.15139/S3/MUSYTQ) published in MEASURE Evaluation Dataverse (view at https://dataverse.unc.edu/dataverse/MEP3). Anonymized transcripts are made available to researchers at no charge. 

Response to reviewers

Reviewers' comments:

Reviewer's Responses to Questions

Comments to the Author

1. Is the manuscript technically sound, and do the data support the conclusions?

Reviewer #1: Yes

Reviewer #2: Yes

No response from the authors is required.

2. Has the statistical analysis been performed appropriately and rigorously? 

Reviewer #1: Yes

Reviewer #2: N/A

 No response from the authors is required.

3. Have the authors made all data underlying the findings in their manuscript fully available?

Reviewer #1: Yes

Reviewer #2: Yes

 No response from the authors is required.

4. Is the manuscript presented in an intelligible fashion and written in standard English?

Reviewer #1: Yes

Reviewer #2: Yes

 No response from the authors is required.

5. Review Comments to the Author

Reviewer #1: A technically sound and academically well written manuscript. The qualitative data analysis was comprehensively explained. However, mention of inter rater reliability by comparing the coding and thematic analysis with co researchers and participants would have further strengthened the analysis. Nevertheless, the manuscript highlights relevant themes pertaining to adherence to TB treatment. Multiple drug resistant strains are a major public health concern of present times.

Response: Thank you for your comment. We agree that calculation of inter-rater reliability would have further strengthened the analysis, however, we applied different techniques to validate the results of the analysis. Two researchers read all study transcripts to ensure that presented results reflect respondents’ views. The lead data analyst worked on coding and shared the code reports with another researcher for input. Both researchers reviewed the code reports, identified sub-themes in each code, and examined the evidence supporting the themes and sub-themes independently and then formed a consensus. Essential concepts and relationships between the different themes and sub-themes were formed. Both researchers worked on synthesizing data and presenting findings in the report.

To address this comment from the reviewer, we provided additional information in the analysis section of the manuscript. 

This is the original version:

We reviewed the code reports, identified sub-themes in each code, and examined the evidence supporting the themes and sub-themes. Essential concepts and relationships between the different themes and sub-themes were formed. Data were synthesized, and findings communicated through the process of writing up and presenting the data, using direct quotes to support the themes. 

This is the modified version: 

Two researchers read all study transcripts and reviewed the code reports, identified sub-themes in each code, examined the evidence supporting the themes and sub-themes independently and then formed a consensus. Essential concepts and relationships between the different themes and sub-themes were formed. Data were synthesized, and findings communicated through the process of writing up and presenting the data, using direct quotes to support the themes. 

Reviewer #2: The manuscript is written in tangible and organized way and touched critical area that should be addressed. However, some printing errors need revision. e.g. line 196, line 270

 Response: Thank you for your comment. We reviewed and revised the manuscript. We fixed errors in lines 196 and 270 as well as other errors related to translation of quotes from Ukrainian to English. 

6. PLOS authors have the option to publish the peer review history of their article (what does this mean?). If published, this will include your full peer review and any attached files.

Do you want your identity to be public for this peer review? For information about this choice, including consent withdrawal, please see our Privacy Policy.

Reviewer #1: No

Reviewer #2: No

No response from the authors is required.

---

## [Editor Report · Decision Letter 1]

14 Aug 2019

What works best for ensuring treatment adherence. Lessons from a social support program for people treated for tuberculosis in Ukraine

PONE-D-19-15537R1

Dear Dr. Charyeva,

We are pleased to inform you that your manuscript has been judged scientifically suitable for publication and will be formally accepted for publication once it complies with all outstanding technical requirements.

Shortly after the formal acceptance letter is sent, an invoice for payment will follow. To ensure an efficient production and billing process, please log <gwmw class="ginger-module-highlighter-mistake-type-1" id="gwmw-15656801839781339269092">into</gwmw> Editorial Manager at https://www.editorialmanager.com/pone/, click the "Update My Information" link at the top of the page, and update your user information. If you have any billing related questions, please contact our Author Billing department directly at authorbilling@plos.org.

With kind regards,

Kahabi Ganka Isangula, MD, MPH, PhD

Academic Editor

PLOS ONE

Additional Editor Comments (optional):

Reviewers' comments:

<gdiv></gdiv>

---

## [Editor Report · Acceptance letter]

19 Aug 2019

PONE-D-19-15537R1 

What works best for ensuring treatment adherence. Lessons from a social support program for people treated for tuberculosis in Ukraine 

Dear Dr. Charyeva:

I am pleased to inform you that your manuscript has been deemed suitable for publication in PLOS ONE. Congratulations! Your manuscript is now with our production department. 

With kind regards,

on behalf of

Dr. Kahabi Ganka Isangula 

Academic Editor

PLOS ONE